

# RMIS-Net: a fast medical image segmentation network based on multilayer perceptron

Binbin Zhang[1], Guoliang Xu[2], Yiying Xing[1], Nanjie Li[2] and Deguang Li[2]

[1] College of Sciences, Shihezi University, Shihezi, China
[2] School of Information Engineering, Luoyang Normal University, Luoyang, China

## ABSTRACT

Medical image segmentation, a pivotal component in diagnostic workflows and therapeutic decision-making, plays a critical role in clinical applications ranging from pathological diagnosis to surgical navigation and treatment evaluation. To address the persistent challenges of computational complexity and efficiency limitations in existing methods, we propose RMIS-Net—an innovative lightweight segmentation network with three core components: a convolutional layer for preliminary feature extraction, a shift-based fully connected layer for parameter-efficient spatial modeling, and a tokenized multilayer perceptron for global context capture. This architecture achieves significant parameter reduction while enhancing local feature representation through optimized shift operations. The network incorporates layer normalization and dropout regularization to ensure training stability, complemented by Gaussian error linear unit (GELU) activation functions for improved non-linear modeling. To further refine segmentation precision, we integrate residual connections for gradient flow optimization, a Dice loss function for class imbalance mitigation, and bilinear interpolation for accurate mask reconstruction. Comprehensive evaluations on two benchmark datasets (2018 Data Science Bowl for cellular structure segmentation and ISIC-2018 for lesion boundary delineation) demonstrate RMIS-Net's superior performance, achieving state-of-the-art metrics including an average F1-score of 0.91 and mean intersection-over-union of 0.82. Remarkably, the proposed architecture requires only 0.03 s per image inference while achieving 27× parameter compression, 10× acceleration in inference speed, and 53× reduction in computational complexity compared to conventional approaches, establishing new benchmarks for efficient yet accurate medical image analysis.

# INTRODUCTION

Image segmentation is a major technology in image processing that involves identifying pixels in an image in order to segment various objectives. The primary function of medical image segmentation is to identify and segment various tissues and organs in medical images, which is critical for disease diagnosis and treatment planning. It is widely used in medical image analysis (*Ma et al., 2021*), pathological diagnosis, surgical planning, treatment monitoring, and other fields. Through medical image segmentation technology,

Corresponding author
Binbin Zhang,
binbinzhanghkj@163.com

it can help doctors understand the scope and distribution of diseases more accurately, thereby helping doctors to make correct diagnosis and treatment plans, improve treatment effects and patients' quality of life.

Compared with ordinary pictures, medical images have more complex imaging methods, due to their noise, low contrast, irregular shape, size and other problems, resulting in the difficulty of segmentation of medical images, and the accuracy and efficiency of segmentation results are also affected. Therefore, how to improve the accuracy, reliability and efficiency of medical image segmentation tasks is still a task with certain challenges and practical application value.

Early medical image segmentation approaches depended heavily on hand-designed features and typical machine learning algorithms, such as threshold segmentation (*Jardim, António & Mora, 2023*), regional growth (*Prabin & Veerappan, 2014*), and boundary segmentation (*Mahmood et al., 2015*). These traditional segmentation methods have certain limitations, requiring a lot of manual labor and expertise, difficulty in adapting to multiple data and scenarios, and not robust enough to problems such as noise and irregular shapes, and cannot achieve the desired segmentation effect for complex medical images.

In recent years, with the successful application of deep learning in many fields, relevant researchers have also proposed many excellent medical image segmentation methods based on deep learning. They can automatically learn features from data, and have strong characterization ability and adaptability, showing better segmentation accuracy and efficiency than traditional medical image segmentation methods. Among them, there is U-Net proposed by *Ronneberger, Fischer & Brox (2015)*, which uses encoder-decoder structure to extract features at different scales, and can effectively fuse high-resolution shallow features and low-resolution deep features, becoming the baseline in the field of medical image segmentation. Since U-Net was proposed, many key extensions have been proposed on this basis, such as UNet++ (*Zhou et al., 2018*), UNet3+ (*Ping & Sheng, 2023*), ResUNet (*Zhang, Liu & Wang, 2018*), 3D UNet (*Pani & Chawla, 2024*), V-Net (*Milletari, Navab & Ahmadi, 2016*), Y-Net (*Mehta et al., 2018*), and KiUNet (*Valanarasu et al., 2020*; *Valanarasu et al., 2021*). In 2021, *Chen et al. (2021)* present a unique TransUNet that combines the advantages of Transformers and U-Net as a powerful alternative to medical imagine segmentation. On the one hand, the Transformer uses tokenized imagine blocks from convolutional neural network (CNN) feature maps as input sequences to extract global context. In contrast, the decoder upsamples the encoded features before combining them with a high-resolution CNN feature map to obtain exact location. Many transformer-based network extensions have now been widely used for medical imagine segmentation, including MedT (*Dosovitskiy et al., 2020*), Swin Transformer (*Cao et al., 2022*), TransBTS (*Wang et al., 2021*), and UNet Transformer (UNETR) (*Hatamizadeh et al., 2022*).

As can be seen from most of the literature above, almost all of the above work is focused on improving network performance, but less concerned with computational complexity, inference time, or number of parameters, which are also critical for many practical applications. Recently, networks based on multilayer perceptrons (MLP) (*Li et al., 2025*;

*Touvron et al., 2022*; *Lian et al., 2021*; *Tolstikhin et al., 2021*) have been found to be competent for computer vision tasks. Particularly, MLP Mixer is a type of MLP-based network with less computational complexity and higher performance. Based on these previous works, this article proposes a fast medical image segmentation network based on MLP, named RMIS-Net, aiming to improve the accuracy and efficiency of medical image segmentation.

The main contributions of this article are as follows: (1) A fast medical image segmentation network based on multilayer perceptron is proposed, which significantly improves inference speed while maintaining accuracy and substantially reduces computational complexity. (2) A shift-based fully connected layer is proposed, and a tokenized multilayer perceptron is used in the latent space to extract local information corresponding to different axial shifts; (3) Successfully improve the performance of medical image segmentation problems with higher inference speed, fewer parameters and lower computational complexity.

The organization of this article is as follows. "Related works" presents related work of medical image segmentation network based on different networks. In "RMIS-Net segmentation networks", we introduces our model named as RMIS-Net, a fast medical image segmentation network designed using multi-layer perceptron, experimental results on the publicly available 2018 Data Science Bowl and ISIC-2018 Lesion Boundary Segmentation datasets are presented in "Experiments", and "Conclusion and Future Works" summarizes the work of this article and looks forward to the future work.

# RELATED WORKS

## Medical image segmentation network based on machine learning approaches

Traditional machine learning approaches for medical image segmentation typically involve feature extraction followed by classification or clustering. These methods rely on handcrafted features, such as texture, intensity, shape, and edge information, which are then fed into machine learning models like support vector machines (SVMs) (*Rahman, Antani & Thoma, 2011*), random forests, k-nearest neighbors (k-NN) (*Ramteke & Monali, 2012*), and Gaussian mixture models (GMMs) (*Greenspan & Pinhas, 2007*). These approaches are often computationally efficient and provide interpretable results, making them suitable for specific medical imaging tasks. SVMs have been applied to segment brain tumors, liver lesions, and lung nodules by leveraging intensity and texture features extracted from medical images. Random forests are ensemble learning methods that combine multiple decision trees to improve segmentation accuracy. They are particularly effective in handling noisy data and have been used for segmenting organs like the prostate, heart, and kidneys. k-NN is a simple yet effective algorithm for segmentation tasks, especially when the data distribution is non-parametric. It has been used for segmenting structures like the hippocampus and breast lesions by comparing the similarity of image patches based on feature vectors. GMMs are probabilistic models that assume data points are generated from a mixture of Gaussian distributions. They have been applied to segment brain tissues, such as gray matter, white matter, and cerebrospinal fluid,
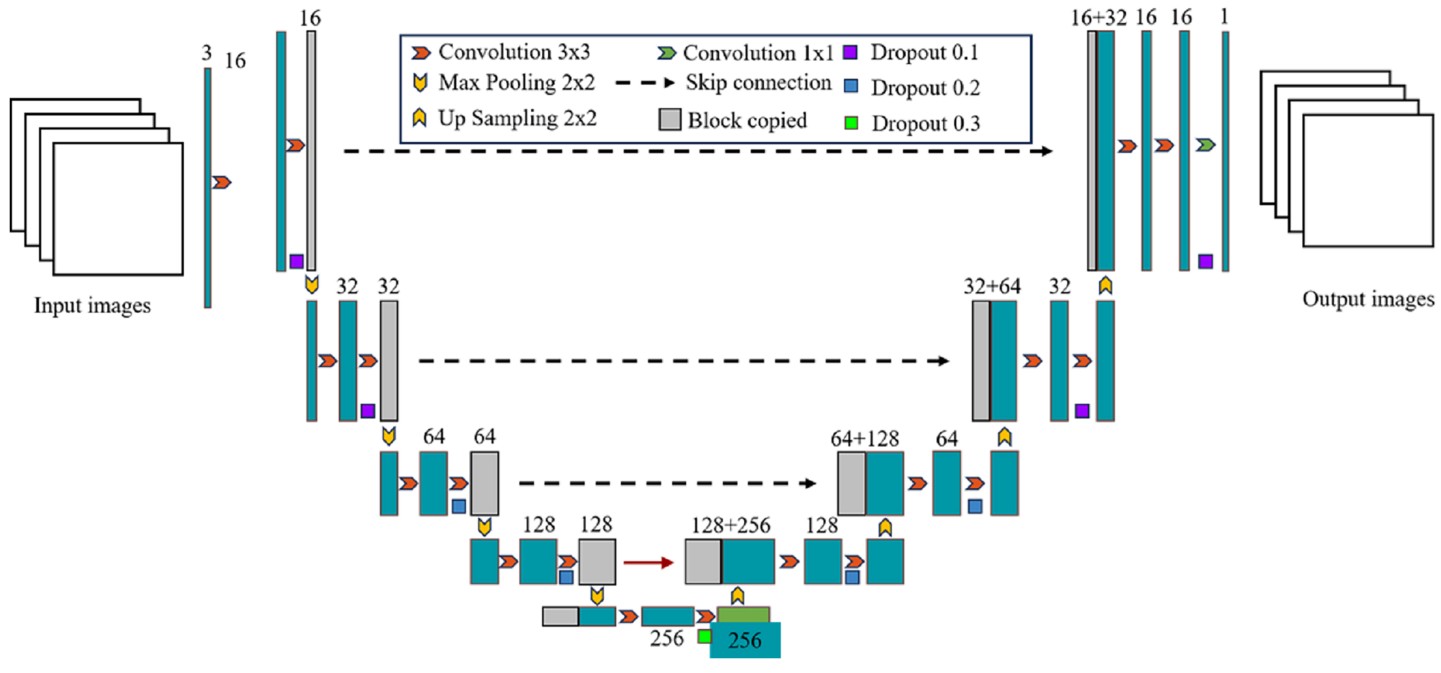

**Figure 1 Detailed overall structure of U-Net network.**

by modeling the intensity distributions of these tissues in MRI scans. Traditional machine learning approaches are often compared to deep learning methods, which have gained popularity due to their ability to automatically learn features from data. While deep learning models have achieved state-of-the-art performance in many segmentation tasks, traditional machine learning methods remain relevant in scenarios where interpretability, computational efficiency, and small datasets are critical.

## Medical image segmentation network based on U-Net

U-Net was originally proposed by *Ronneberger, Fischer & Brox (2015)* and has been widely used in the field of medical image segmentation, such as lung, heart, breast, liver and other medical image segmentation tasks. It is based on the fully convolutional network (FCN) (*Long, Shelhamer & Darrell, 2015*) and uses an encoder-decoder structure. The left encoder part is composed of multiple convolutional layers and pooling layers to extract the features of the input image; The right decoder part consists of multiple deconvolution layers and upsampling layers, which are used to restore the features extracted by the encoder to the original image size and generate segmentation results of the target region; The left and right structures are combined to form a U-shaped structure, which can effectively retain the information in the image. Another feature of the U-Net network is the use of skip connection, which crop the output of the downsampling layer and concatenate it with the corresponding upsampling layer in the channel dimension, so that the high-level semantic features are combined with the low-level spatial information to supplement the detailed information, thereby improving the segmentation accuracy. The U-Net network architecture is shown in Fig. 1. The input data are typically grayscale or color images. The

encoder extracts features from the input image through operations like convolution and pooling, while also reducing the size of the feature map. The decoder restores the size of the feature map through upsampling and convolution, and generates the final segmentation result.

Since the proposal of U-Net network, convolutional neural network based on U-Net has made remarkable achievements in the field of medical image segmentation. Subsequent researchers have made many improvements based on this, mainly including UNet++ (*Zhou et al., 2018*) that modifies the skipping connection, ResUNet (*Zhang, Liu & Wang, 2018*) that introduces residual connection to U-Net, V-Net that introduces 3D convolutional layers and void convolutional layers to adapt to 3D medical image data (*Milletari, Navab & Ahmadi, 2016*), and Y-Net that uses Y-shaped structure to branch encoder and decoder separately and fuse them at the final level (*Mehta et al., 2018*). The segmentation network based on U-Net can perform feature extraction at different scales and has strong feature learning ability, but the extraction of information such as target position and attitude is insufficient, and the local-global relationship cannot be well modeled, and it still has limitations in processing medical image data with strong position correlation.

## Medical image segmentation network based on attention mechanism

The medical image segmentation network based on the attention mechanism is a kind of deep convolutional neural network that uses the attention mechanism to improve the accuracy of medical image segmentation. The main idea of this kind of network is to add an attention mechanism to the network so that the network can focus more accurately on the area of interest, thereby improving the quality of segmentation results. In medical image segmentation, attention mechanisms are often used to guide networks to better extract features in regions of interest and exclude interference from irrelevant regions. The implementation of the medical image segmentation network based on the attention mechanism includes: (1) adding an attention module to the encoder-decoder network. The attention module is usually designed to consist of two parts: attention mechanism and modulation factor. This method calculates different weights depending on the input data, and the modulation factor can further adjust these weights to better suit specific task needs. (2) Adopt the Attention Context Embedding approach. This method can add context information to the network to help the network better understand the background and semantic information of the input data, thereby further improving the segmentation accuracy.

In recent years, relevant researchers have also proposed a series of good medical image segmentation networks based on the attention mechanism. *Oktay et al. (2018)* proposed Attention U-Net in 2018, and the attention module consists of two parts: channel attention and spatial attention. Channel attention calculates the weight coefficient for each channel in the feature map to weight different feature map channels. Channel attention adopts the method of global average pooling, compressing the feature map into a vector containing the weights of each channel, and then mapping the vectors to a set of channel weights through a fully connected layer. Spatial attention calculates the weight coefficient for each

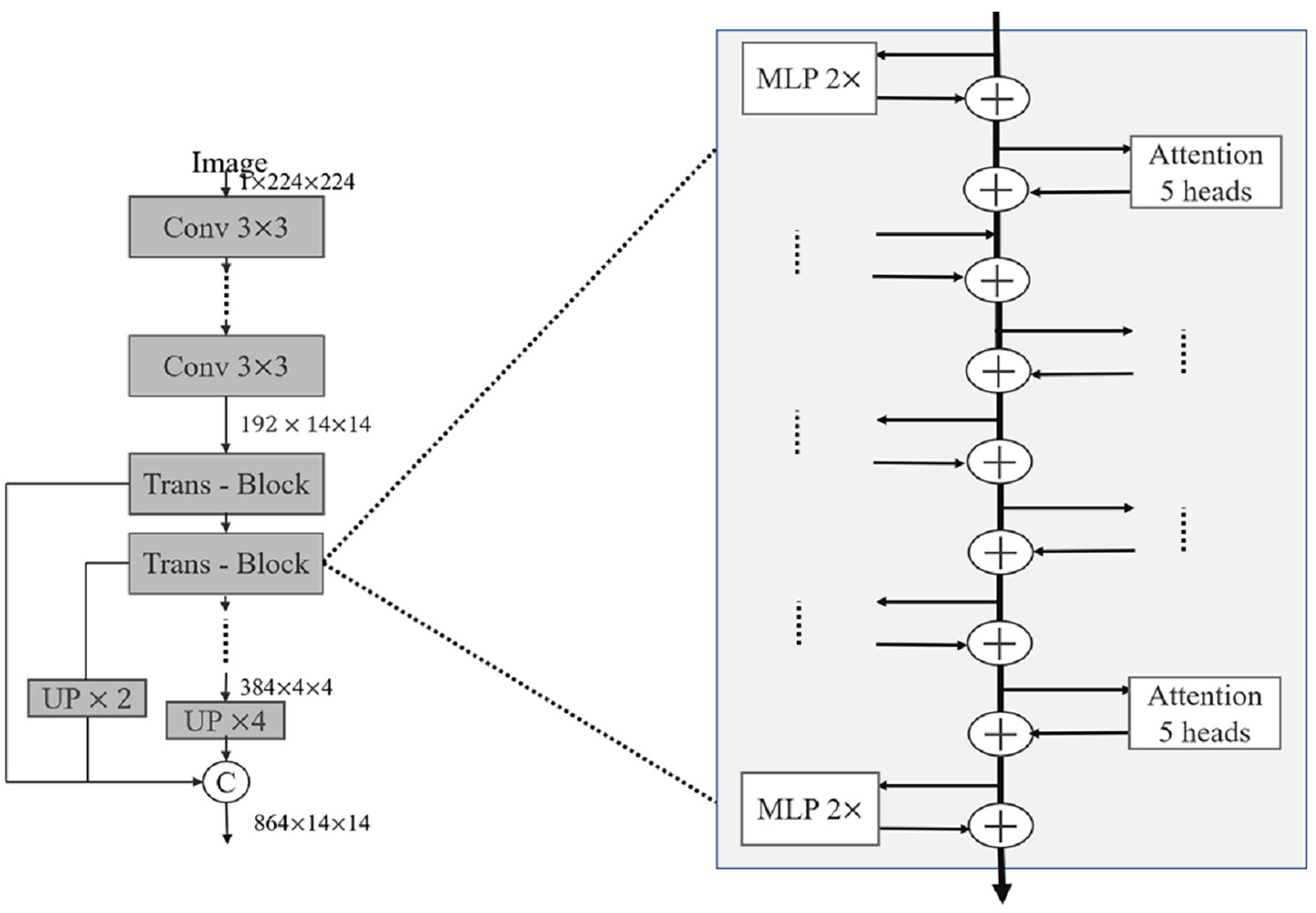

**Figure 2 Detailed internal structure of LeViT block.**               

pixel in the feature map to weight each pixel. Spatial attention consists of two submodules: the spatial transformation network and the channel pooling layer. The spatial transformation network adopts an encoder-decoder structure similar to that in U-Net to downsample and upsample the feature map, and calculate the weight coefficient of each pixel. The channel pooling layer reduces the channel dimension by pooling the feature map globally. The output of the attention module is a feature map of intelligent summation of elements, where the value of each pixel is the weighted sum of its attention weight coefficients on channels and spaces. In 2021, *Xu et al. (2023)* proposed LeViT-UNet, which combines the attention mechanism in ViT and the encoder-decoder structure in U-Net to achieve efficient and accurate medical image segmentation. In each Trans-Block shown in Fig. 2, the self-attention module is first used for feature extraction, then the multilayer perceptron is used for feature mapping, and finally the cross-layer connection technology is used to fuse the feature map of the layer with the feature maps of all previous layers. The decoder adopts the structure of U-Net, in which each decoder layer is connected to the

corresponding layer in the encoder. In addition, attention mechanisms and skip connection techniques are used to help the network better capture local and global information in images.

## Medical image segmentation network based on transformer

Transformer-based medical image segmentation network is an emerging medical image segmentation method, which is developed on the basis of the widely used Transformer model in the field of natural language processing. This method utilizes the self-attention mechanism of the Transformer to process spatial information in medical images, and learns feature representations through the multi-head self-attention mechanism and feed-forward network. The main idea of the method is to convert medical images into a sequence and then input them into a Transformer model for processing. Specifically, the Transformer-based medical image segmentation network includes an encoder and a decoder. The encoder consists of multiple transformer blocks, each of which includes a feed-forward network and a multi-head self-attention mechanism. The decoder also consists of multiple transformer blocks, each of which includes a feed-forward network, a multi-head self-attention mechanism and an attention mechanism. In each transformer block, the information of the encoder and decoder is fused by the multi-head self-attention mechanism and the attention mechanism to generate segmentation results. The Transformer model has good spatial modeling capabilities, which can better process the spatial information in medical images, and has fewer parameters and higher efficiency, which can learn better feature representations in less time, thereby improving the efficiency and accuracy of medical image segmentation.

At present, relevant researchers have also proposed some good medical image segmentation networks based on Transformer. *Cao et al. (2022)* proposed Swin-Unet in 2021 as shown in Fig. 3, which uses a combination of Transformer module and Unet structure to perform medical image segmentation tasks. The encoder part of Swin-Unet adopts the basic structure of Swin Transformer, including Swin Block and Swin Layer, which are used to extract the features of the input image. In the decoder section, Swin-Unet uses a Unet-like structure for upsampling and feature fusion to produce the final segmentation result. A large number of experiments on multi-organ and cardiac segmentation datasets show that the proposed method has robust generalization ability and better segmentation accuracy. *Chen et al. (2021)* proposed TransUNet in 2021, a network that adaptively learns feature representations of images using a self-attention mechanism. Its network structure is mainly divided into two parts: Encoder and Decoder. The Encoder part uses the Transformer module instead of the traditional convolutional neural network to divide the input image into multiple subregions, and then perform self-attention calculations on each subregion to generate region-level feature representations. The Decoder part uses a conventional convolutional neural network structure to decode the features of the encoder output to generate the final segmentation result. Detail experiments on multiple medical image segmentation datasets show that the TransUNet network could obtain better segmentation effects than traditional

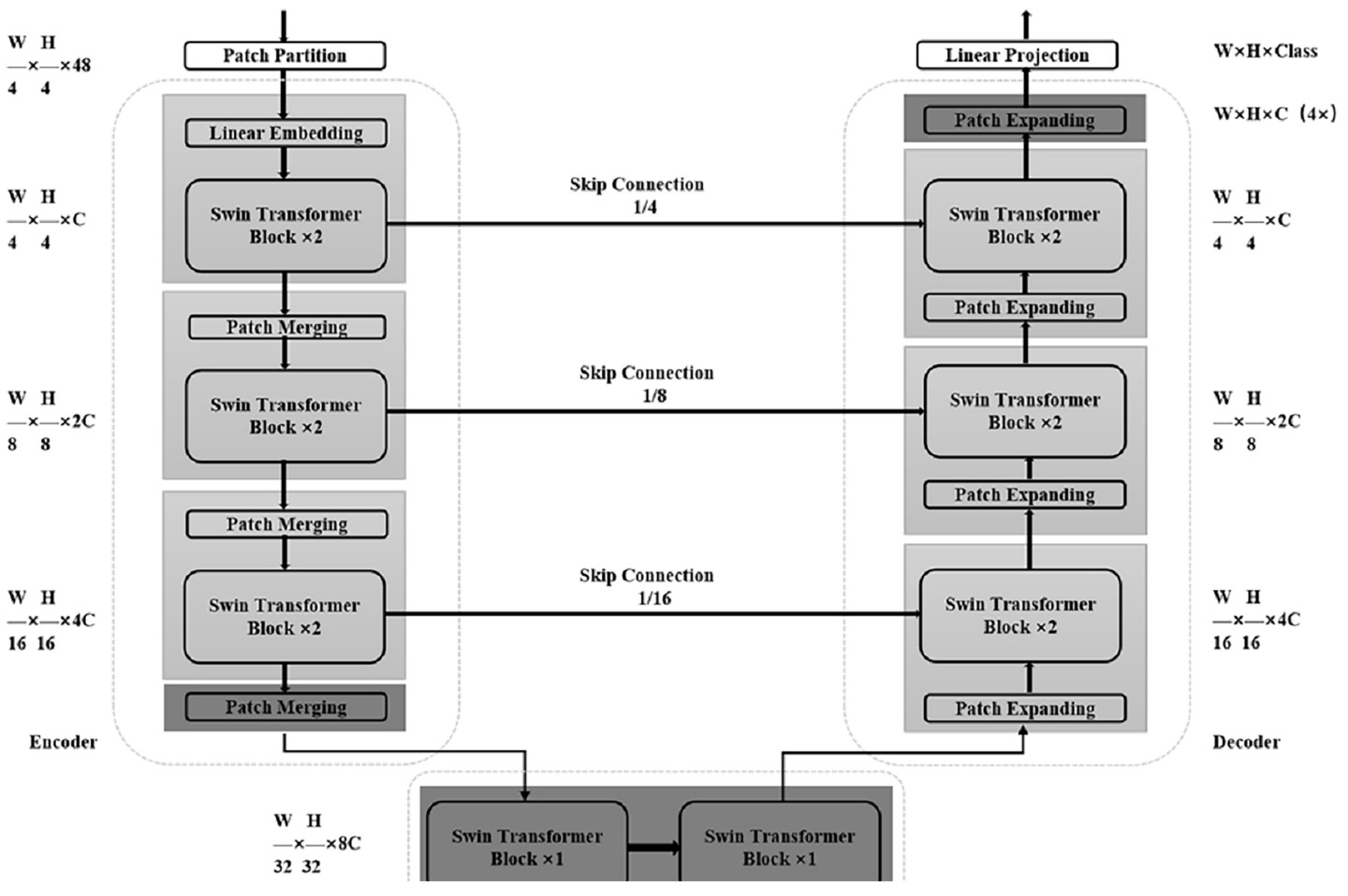

**Figure 3  Detailed overall structure of Swin-Unet.**               

convolutional neural network-based methods, while also having better interpretability and fewer parameters.

Traditional machine learning methods typically rely on handcrafted features, which are well-suited for small datasets, and offer high computational efficiency. However, they depend heavily on the developer's expertise for feature design, have limited generalization capabilities, and often underperform on complex medical images compared to deep learning methods. U-Net can automatically learn features, and its encoder-decoder architecture combined with skip connections enables effective multi-scale information capture and feature fusion. Nevertheless, U-Net requires a large amount of data and may overfit when data is insufficient. Additionally, its ability to model long-range dependencies is limited, as it primarily relies on local convolutional operations, making it difficult to capture global contextual information in images. Attention mechanisms can capture long-range dependencies in images, improving segmentation accuracy, and can be flexibly integrated into existing networks (*e.g.*, U-Net) to enhance the model's focus on important features. However, attention mechanisms increase computational overhead, especially for high-resolution images, and are challenging to train while offering poor interpretability.

Transformers, capable of naturally handling multi-modal data (*e.g.*, CT, MRI, PET), are well-suited for complex medical image analysis tasks. However, they require significant computational resources and memory, resulting in high training and inference costs, as well as a large demand for data and limited interpretability. The proposed fast medical image segmentation network based on multi-layer perceptrons (MLPs) in this article aims to address the issues of high complexity and low efficiency in medical image segmentation. The model consists of convolutional layers and shift-based fully connected layers, and it employs tokenized MLPs in the latent space to reduce the number of parameters and computational complexity, our approach maximizes network performance while ensuring model accuracy.

# RMIS-NET SEGMENTATION NETWORKS

## Overall architecture

This article designs the RMIS-Net network as a two-stage encoder-decoder architecture that includes two stages: the convolution stage and the tokenized multilayer perceptron stage. Figure 4 shows RMIS-Net, a network architecture for medical image segmentation proposed in this article. In the encoder part, the first three layers of the network are convolutional blocks, and the last two layers are tokenized multilayer perceptron blocks, which reduce the resolution of the feature map by a factor of two. In the decoder part, the first two layers of the network are tokenized multilayer perceptron blocks, and the last three layers are convolutional blocks, which increase the resolution of the feature map by a factor of two. A skip connection is used between the encoder and decoder, and the number of channels is a hyperparameter expressed as C1 to C5. In the experiments in this article, the number of channels per block in the RMIS-Net network architecture used is set to C1 = 32, C2 = 64, C3 = 128, C4 = 160, C5 = 256 by default. This setting can be adjusted based on actual needs and experimental results to achieve the best performance and results.

RMIS-Net takes convolutional blocks with a smaller number of filters as the initial and final blocks of the network, and uses a novel tokenized multilayer perceptron in the bottleneck that is able to model good representations while maintaining less computation. This article also introduces shift operations in multilayer perceptrons to extract local information corresponding to different axial shifts. Due to the smaller size of tokenized features and the smaller complexity of multilayer perceptrons than convolutional or self-attention and transformers. As a result, we were able to significantly reduce the number of parameters and computational complexity while maintaining good performance. RMIS-Net also adopts techniques such as layer normalization and discarded regularization to avoid overfitting to achieve more stability during training while having better generalization performance during the testing stage. In order to accelerate the convergence of the model and improve the segmentation accuracy of the model, this article introduces residual connection to solve the gradient vanishing problem in deep networks, and the Dice loss function to better deal with the category imbalance problem.
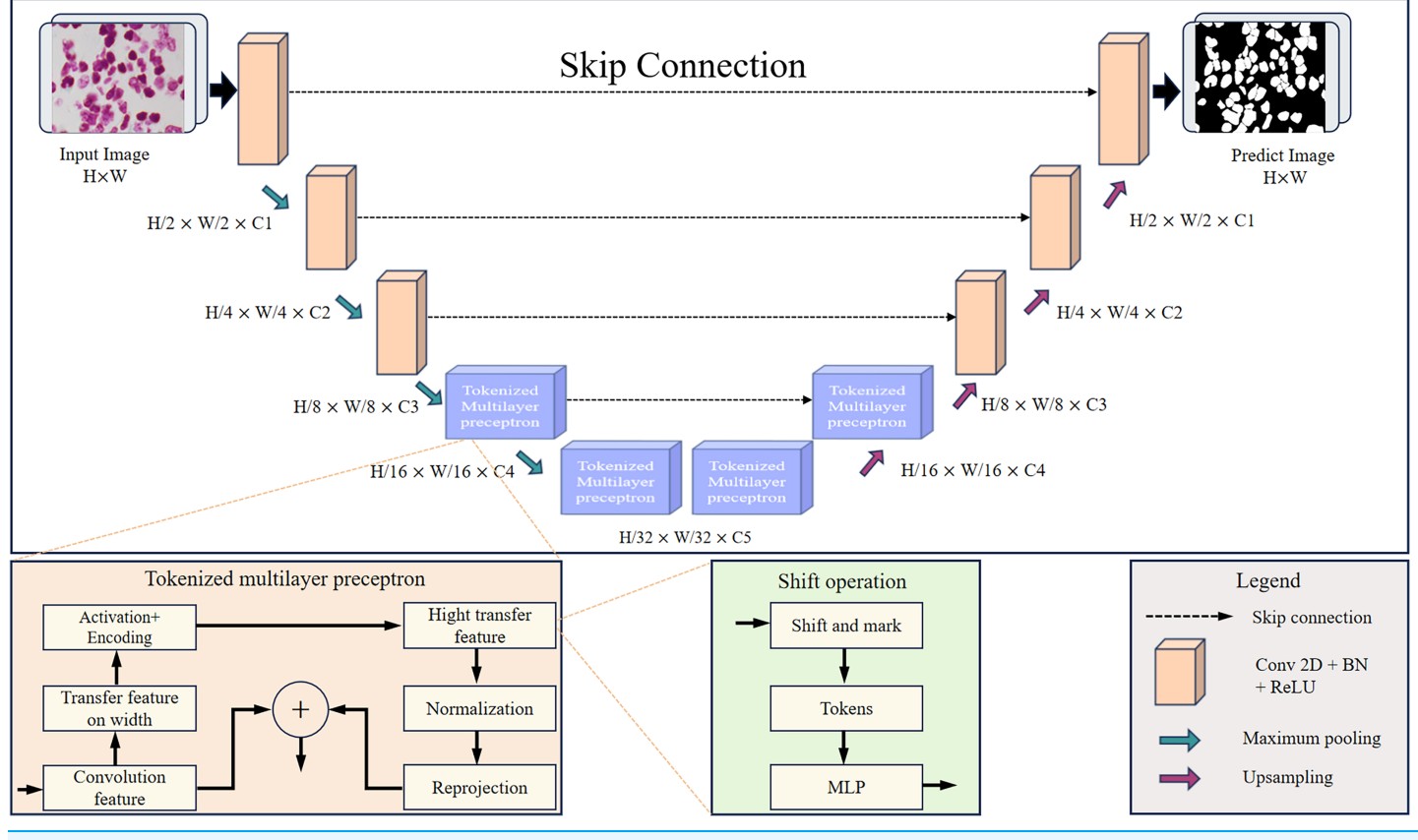

**Figure 4 Detailed overall structure of RMIS-Net segmentation network.**

## Convolution stage

In the convolution stage, a series of effective techniques are used to achieve high-quality image super-resolution reconstruction, and excellent performance and results are achieved in experiments. The proposed method uses a convolutional block structure that contains three main components: the convolutional layer, the batch normalization layer, and the ReLU activation function. The kernel size of the convolutional layer is set to $3 \times 3$, the step size is 1, and the fill is 1, which can effectively preserve the details and spatial information of the image while reducing the impact on the size of the feature map. Batch normalization is a common regularization method that can improve the training stability and convergence speed of the model. The ReLU function enables the output of neurons to be spars, thereby reducing the number of parameters and the complexity of calculations. In the encoder part, a maximum pooling layer with a size of $2 \times 2$ is added for downsampling, which can effectively reduce the size of the feature map and speed up the calculation speed of the model, so as to improve the generalization performance of the model. In the decoder section, bilinear interpolation is used for upsampling of feature maps. Bilinear interpolation (calculated as in Eq. (1)) is a common interpolation method that can restore the details and spatial information of the image by weighting the values of neighboring

pixels by calculating the weight of each pixel on the input feature map, while also reducing noise and discontinuities in the image. Compared with the traditional transposed convolutional layer, this method can upsample low-resolution feature maps to high resolution without introducing additional learnable parameters, thereby avoiding the overfitting problem, while not only maintaining the smoothness and continuity of the image, but also avoiding jagged artifacts. In addition to this, this article optimizes the training and performance of the model by using residual connections. Residual connection is a special connection method that models the difference between input and output and adds it to the learning objective of the model, thereby reducing gradient vanishing and gradient explosion problems, while also speeding up the training speed and convergence speed of the model.

$$
\begin{aligned}
g(u, v) = {} & (1 - u)(1 - v) \times f(x, y) + u(1 - v) \times f(x + 1, y) \\
& + v(1 - u) \times f(x, y + 1) + uv \times f(x + 1, y + 1).
\end{aligned}
\tag{1}
$$

Among them, $f(x, y)$ is the pixel value before upsampling of the original image, $g(u, v)$ is the pixel value after upsampling, $(u, v)$ is the pixel position after upsampling, $(x, y)$ is the original pixel position, $u$ and $v$ are the interpolation weights in the range of [0, 1], and $(x, y)$, $(x + 1, y)$, $(x, y + 1)$ and $(x + 1, y + 1)$ are the pixel values of the nearest four pixels.

## Tokenized multilayer perceptron stage

The tokenized multilayer perceptron (tokenized MLP) stage of this article combines multiple processing steps to extract and encode feature information from the image.

### *Shifted multilayer perceptron*

In machine learning and deep learning, a multilayer perceptron is a common neural network structure used to extract complex nonlinear relationships from input features. An important component of the multilayer perceptron is the fully connected layer, where each neuron is connected to all neurons in the previous layer. This fully connected structure faces some problems when processing high-dimensional input data, such as requiring a large number of parameters and calculations, and is also prone to overfitting. To address these issues, this article uses shift operations to improve the performance of multilayer perceptrons. Specifically, this article shifts features on the axis of the shift channel before marking. This practice not only helps the multilayer perceptron to pay more attention to the local area when processing features, but also helps the network better handle high-dimensional data, thereby improving the performance of the model.

In this article, multiple multilayer perceptrons are used in the tokenized multilayer perceptron block to process features. The specific method is to introduce multi-scale feature information by shifting width up and height shift on different multilayer perceptrons, so that the network can pay attention to features of different scales at the same time. This method can help us create random windows and introduce location, so that the network can make full use of the features of different locations. The approach in this article

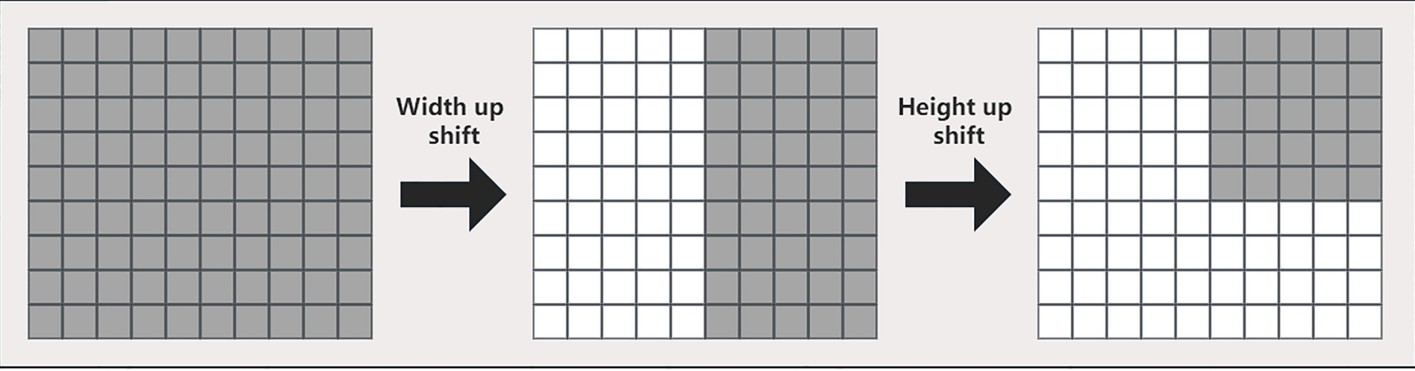

**Figure 5 Detailed width and height shift operation of the input image.**

is similar to that in axial attention (*Wang et al., 2020*), but we use a different implementation, as shown in Fig. 5. First, each marked feature is split into n different zones and moved to the mth position according to the pre-specified axis line. For each partition, we use a different shift offset in each multilayer perceptron to capture features at different locations. This multi-scale processing method enables the network in this article to pay attention to the features of different scales at the same time, so as to better capture the spatial information of the image. Then, by shifting the features before marking, the multilayer perceptron can pay more attention to the local area when processing the features, so that the network can better capture the local feature information and improve the location of the network, thereby improving the model performance.

### Depthwise convolutional layer

Features processed by multilayer perceptrons are passed to the depthwise convolutional layer (DWConv) (*Xie et al., 2021*) for convolution operations. DWConv is a depthwise separable convolution that encodes the location information of multilayer perceptron features, helping the network better understand the spatial structure in the image, which is shown in Fig. 6. It uses a convolution kernel size of 3 × 3 to provide a local receptive field, so that each convolution kernel can capture local feature information in the image. At the same time, to convert the feature information into markers and prepare them for subsequent processing, the number of channels is set to the embedding dimension E. In convolutional layers, the 3 × 3 convolutional kernel only needs to multiply the input's 3 × 3 neighborhood pixel values, so the required computation is relatively small. The embedding dimension can be controlled by adjusting the number of channels, which can reduce computational and memory consumption while ensuring model performance.

Compared with the traditional convolution operation, DWConv separates the two processes of spatial convolution and channel convolution, which first performs spatial convolution and then channel convolution. The spatial convolution operation in DWConv uses a fixed convolution kernel, while the channel convolution operation uses matrix multiplication to obtain better spatial structure features by convoluting the input channel by channel. While maintaining a relatively small number of model parameters, DWConv

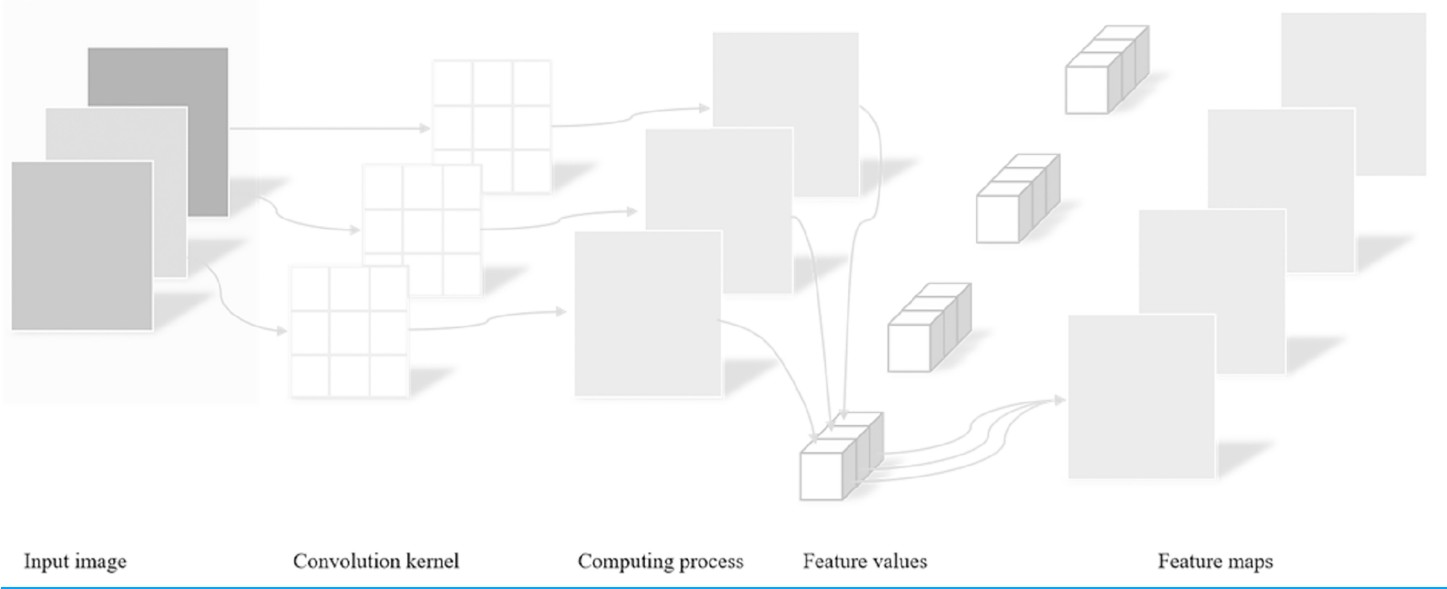

| Input image | Convolution kernel | Computing process | Feature values | Feature maps |

**Figure 6 Detailed process of depthwise separable convolution.**

can not only improve the computational efficiency of the model, but also improve the expressiveness of the model.

### Smooth activation function

Gaussian error linear unit (GELU) (*Hendrycks & Gimpel, 2016*) has been widely used in advanced image processing architectures, so GELU was chosen as the activation function in this article. It is a smoother activation function that helps maintain the smoothness and continuity of the network. Compared with the ReLU activation function, GELU has a certain output value in the negative region, which can help the network better adapt to the negative input and maintain a smoother activation output, so it is more stable in the gradient calculation in backpropagation. In addition, the function not only helps to better optimize network parameters, but also improves the performance of the model. The definition of GELU is shown in Eq. (2):

$$GELU(x) = x \times \phi(x) = x \times \left(1 + erf\left(\frac{x}{\sqrt{2}}\right)\right) \times 0.5 \tag{2}$$

where $erf\left(\frac{x}{\sqrt{2}}\right)$ is the Gaussian error function and $\phi(x)$ is the cumulative distribution function.

### Layer normalization

In the tokenized multilayer perceptron module, this article further transforms the features through a shift operation and adds the original marks to the converted features through residual connections. Next, layer normalization (LN) normalizes the output features. Compared with batch normalization (BN), it has the advantage that it is not affected by

batch size and is suitable for various scenarios. Layer normalization is normalization on each feature of each sample, that is, normalization in the labeled dimension. It is calculated similarly to batch normalization, but when calculating the mean and variance, it is calculated separately for each feature of each sample, rather than for a feature of all samples for the entire batch. It enables better handling of differences between different samples, thereby improving the robustness of the model. Layer normalization can also speed up the training process by mitigating the vanishing gradient problem in deep neural networks. Compared to batch normalization, it can be more easily applied to larger models and datasets because it does not need to account for batch size resulting in a smaller computation amount. In the tokenized multilayer perceptron stage, there may be differences between different labels, and normalization along the marker dimension can better handle this difference. Therefore, in this stage, layer normalization is chosen as the normalization method for this article and is used to pass the output features to the next block. The calculation formula in the tokenized multilayer perceptron stage is as follows:

$$X_{shift} = Shift_W(X); T_W = Tokenize(X_{shift}) \tag{3}$$

$$Y = f(DWConv(MLP(T_W))) \tag{4}$$

$$Y_{shift} = Shift_H(Y); T_H = Tokenize(Y_{shift}) \tag{5}$$

$$Y = f(LN(T + MLP(GELU(T_H)))) \tag{6}$$

where $T$ represents markup, $H$ represents height, $W$ represents width, DWConv represents depthwise convolution, and $LN$ represents layer normalization. It should be noted that the above calculations are performed on the hidden layer dimension $H$, which is strictly smaller than the dimension $\frac{H}{N} \times \frac{H}{N}$ of the feature map, and $N$ is a multiple of 2. In this design, the calculation and parameters of the Tokenized MLP block are mainly focused on the embedding dimension, rather than the dimension of the entire feature map, which reduces the computational complexity and the amount of parameters, improves the computational efficiency, and balances the model performance and efficiency.

# EXPERIMENTS

## Datasets and data preprocessing

### Datasets

In this article, the performance of the proposed RMIS-Net segmentation network on different datasets and different disease types is comprehensively evaluated on two publicly available medical image datasets, the 2018 Data Science Bowl dataset (https://www.kaggle.com/c/data-science-bowl-2018/) and the ISIC-2018 Lesion Boundary Segmentation dataset (https://challenge.isic-archive.com/landing/2018/45/).

The 2018 Data Science Bowl dataset is provided by the National Cancer Institute (NCI) and the National Center for Computer and Applied Mathematics (NCCM). It contains 2,883 CT scan images of lung nodules with variable resolution. Figure 7 shows a CT scan image of a lung nodule and an example of a segmentation mask, which were annotated by a

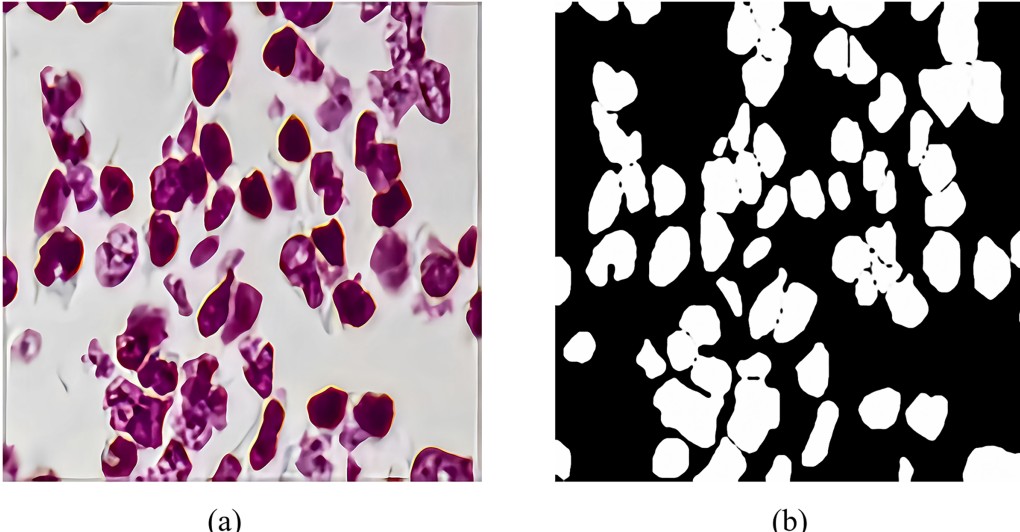

(a)                                              (b)

**Figure 7 The feature generation of target segmentation in the CT scan image.** (A) The CT scan images of pulmonary nodules; (B) shows the segmentation masks of pulmonary nodules.

professional physician and used to evaluate the performance of RMIS-Net on the task of segmentation of lung nodules. Its purpose is to encourage data scientists and medical professionals to collaborate to develop more accurate and efficient lung nodule segmentation algorithms, thereby improving the rate of early diagnosis of lung cancer. The emergence of this dataset is very valuable for the research and practice of medical image analysis.

Input Images: Input images serve as the starting point of the entire processing workflow. In this study, color images are used as inputs for subsequent analysis and recognition tasks. These images are represented as matrices of pixels, where each pixel contains color information comprising three channels: red, green, and blue.

Convolution kernel: A convolution kernel is a small matrix applied to the input image with the purpose of performing convolution operations on the image to extract useful features such as edges, textures, *etc*. The size and weights of the convolution kernel can be pre-defined or learned through training. Different convolution kernels are capable of capturing various types of image features.

Computing process: The computing process refers to sliding the convolution kernel over the input image and performing dot product operations at every position followed by summing these products. This process essentially involves mathematical operations between the convolution kernel and local regions of the image, aiding in extracting high-level features from the original image. By adjusting the position of the convolution kernel, the entire image can be scanned to extract global information.

Feature values: After completing the aforementioned convolution operation, a series of numerical values are obtained, which reflect the feature intensity of specific areas in the

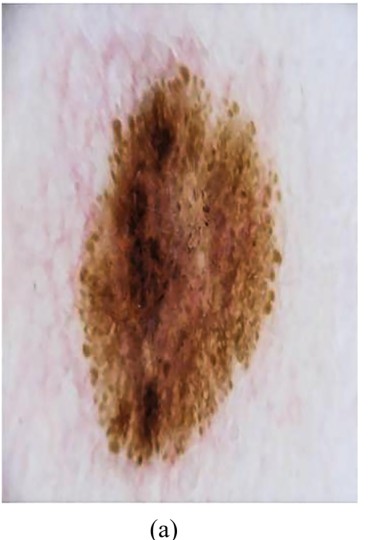 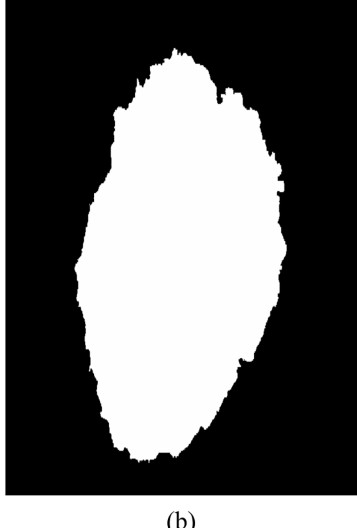

(a)  (b)

**Figure 8 Examples of dermoscopic images and segmentation masks.** (A) Dermoscopic images. (B) The segmentation masks of the image.

image. These numerical values are known as feature values, describing the manifestation of features at different positions within the input image. For instance, in edge detection, feature values may indicate the presence and strength of an edge within a specific area.

Feature maps: Feature maps are new images generated after convolution operations, showcasing the results of the original image processed by the convolution kernel. Each feature map highlights different aspects or features of the input image, such as edges in certain directions, corners, or other shape features. In deep learning networks, multiple convolutional layers produce a series of feature maps. As the depth of the network increases, these feature maps transition from low-level features (like edges) to high-level features (such as parts or overall structures of objects).

The ISIC-2018 Lesion Boundary Segmentation dataset contains various types of skin lesions such as melanoma, benign melanoma nevi, basal cell carcinoma and keratinoma, with a total of 10,015 high-resolution dermoscopic images with different resolutions and certain noise and distortion. Figure 8 shows an example of dermoscopy images and segmentation masks, both labeled by a physician, that were used to evaluate the performance of RMIS-Net on the task of segmentation of skin lesions.

The above two datasets have rich image quantity and reference standards labeled by professional doctors, which can effectively evaluate the accuracy, robustness and generalization ability of the methods, and objectively quantify and compare the research results. Experimental evaluation on these challenging datasets not only helps to ensure the reliability and reproducibility of experimental results, but also provides a better understanding of the performance and application prospects of the proposed model in practical medical applications.

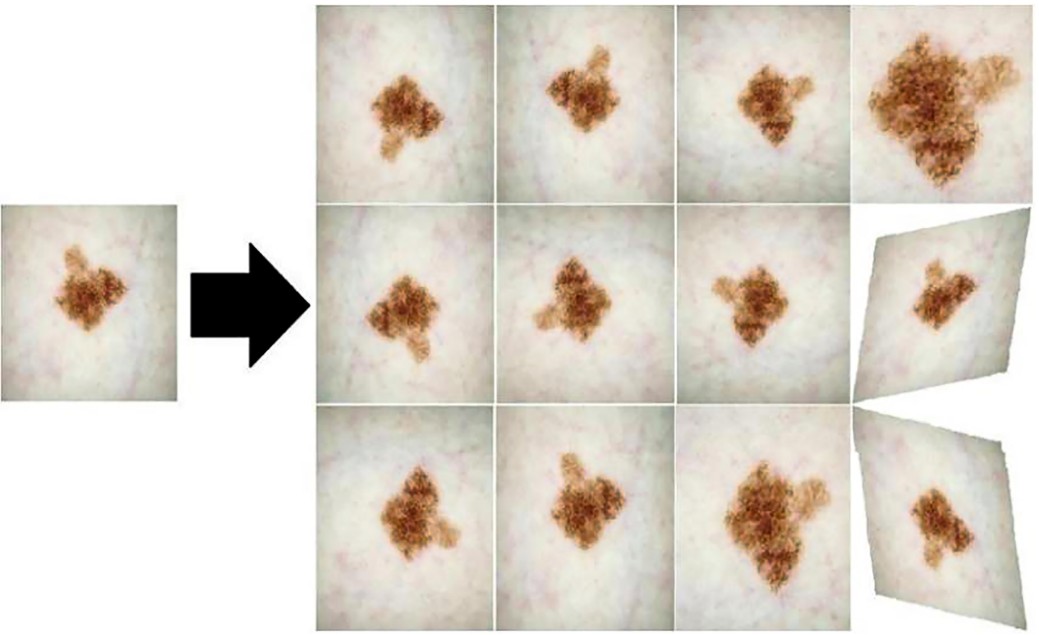

**Figure 9 Three data enhancement operations: horizontal flipping, rotation, and cutting.**

## Data preprocessing

This article preprocesses the dataset as follows:

(1) In order to unify the processing and reduce the computational complexity, the image size of the 2018 Data Science Bowl dataset and the ISIC 2018 dataset is adjusted to 256 × 256 and 512 × 512, respectively. This action not only helps maintain the consistency of the input images, but also complies with the design and parameter settings of the medical image segmentation network in this article. In addition, this operation improves the reproducibility of experiments and comparability of results while maintaining image information.

(2) Data augmentation. In order to solve the problem that the medical image segmentation model may be limited by a limited number of available samples in the training stage, which may lead to the overfitting phenomenon of the model, this article adopts the data augmentation method to expand the diversity of the samples for the input samples, so as to mitigate the negative impact of the overfitting phenomenon on the model performance, and then improve the robustness and generalization ability of the model. Figure 9 shows the three data augmentation operations of horizontal flipping, rotation, and cutting randomly applied in this article, each of which is applied to the training set of each dataset with a probability of 0.20. These operations can introduce image variations for different angles, positions, and orientations, helping the model better adapt to different image inputs.

**Table 1 Details of the medical segmentation dataset used in this article.**

| Dataset name | Number of images | Training set | Validation set | Test set |
|---|---|---|---|---|
| 2018 Data Science Bowl | 2,883 | 2,308 | 431 | 144 |
| ISIC 2018 | 10,015 | 8,015 | 1,600 | 400 |

(3) Data splitting. For the study of medical image analysis, the splitting scheme of the dataset is directly related to the generalization ability and performance of the model. Therefore, data splitting requires special attention, taking into account factors such as dataset size, sample distribution, labeling quality, and case diversity. Details of the two medical image segmentation datasets used in this article are listed in Table 1, including the number of images, the division of the training set, the validation set, and the test set. This article adopts a common random splitting method to avoid specific biases and structural differences in the dataset. At the same time, special consideration is given to factors such as sample distribution and case diversity to ensure that the model can be fully trained and tested on each subset, thereby improving the generalization ability of the model.

## Evaluation metrics

Where accuracy measures the overall correctness of the predictions. Precision measures the proportion of true positive predictions among all positive predictions. Recall measures the proportion of actual positives that are correctly identified. F1 provides a harmonic mean of precision and recall, offering a single measure to evaluate model performance. IoU measures the overlap between the predicted and ground truth regions, often used to assess segmentation quality.

$$Accuracy = \frac{TP + TN}{TP + TN + FP + FN} \tag{7}$$

$$Precision = \frac{TP}{TP + FP} \tag{8}$$

$$Recall = \frac{TP}{TP + FN} \tag{9}$$

$$F_1 = \frac{2 \times Precision \times Recall}{Precision + Recall} \tag{10}$$

$$IoU = \frac{TP}{TP + FP + FN}. \tag{11}$$

In order to comprehensively compare the performance of RMIS-Net and other popular models, this article uses the evaluation metrics model parameter number, inference speed, GFLOPs (billion floating point operations), F1 score, and intersection over union (IoU) (cross-union ratio) to evaluate. By calculating the mean and variance of these metrics, we can comprehensively evaluate the performance of RMIS-Net and other models in different aspects to better understand their performance differences on datasets relevant for clinical diagnosis.

**Table 2 Software and hardware configuration used by RMIS-Net.**

| Hardware and software names | Parameter |
| --- | --- |
| SSDs | 1 TB |
| Memory | 128 GB |
| Nvidia GeForce RTX 3090 GPU | 24 G |
| Intel CPU | 128 G DDR4 |
| Ubuntu | 18.04.6 |
| Python | 3.7 |
| Torch | 1.10.1 |
| CUDA | 11.3 |

## Experimental platform

RMIS-Net is to build the programming environment of the deep learning framework Pytorch on the Ubuntu 18.04.06 system to conduct all relevant experiments, and the software and hardware environment settings used are shown in Table 2.

## Experiment setup

During training, the Adam optimizer with an initial learning rate of 0.0001 and a momentum of 0.9 is used as the optimization algorithm of the model. The training is carried out with a batch size of 8 and a training period of 500, and an early stop mechanism is used in the process, and the training is stopped when the validation set loss value of 2 consecutive epochs no longer decreases. In addition, during the training process, this article also uses the cosine annealing learning rate scheduler to calculate the current learning rate, and then applies it to the weight parameter used to update the model in the optimizer, and its calculation formula is shown in Eq. (12):

$$lr = lr_{\min} + 0.5 \times (lr_{\max} - lr_{\min}) \times (1 + \cos(epoch/T_{\max} \times pi)) \qquad (12)$$

where $lr$ is the current learning rate, $lr_{\min}$ is the minimum learning rate, $lr_{\max}$ is the maximum learning rate, $epoch$ is the number of current training cycles, $T_{\max}$ is the maximum number of training cycles, and $pi$ is $\pi$. By adjusting $lr_{\min}$, $lr_{\max}$ and $T_{\max}$, the speed and amplitude of the learning rate can be flexibly controlled, so as to optimize the training process of the model.

## Loss function

In this article, a combination of binary cross-entropy (BCE) and Dice loss is used to train the proposed model RMIS-Net during training. The loss function definition for the entire network model is shown in Eq. (13):

$$
\begin{aligned}
L &= 0.5BCE(\hat{y}, y) + Dice(\hat{y}, y) \\
&= 0.5BCE(\hat{y}, y) + \frac{(2 \times \sum(\hat{y}_i \times y_i) + \varepsilon)}{(\sum \hat{y}_i + \sum y_i + \varepsilon)}
\end{aligned}
\qquad (13)
$$

**Table 3 Performance comparison results under different channel count configurations.**

| Model name | C1 | C2 | C3 | C4 | C5 | Model parameters (M) | Inference time (ms) | GFLOPs | F1 | IoU |
|---|---|---|---|---|---|---|---|---|---|---|
| RMIS-Net | 32 | 64 | 128 | 256 | 512 | 6.74 | 114 | 1.84 | 0.918 | 0.837 |
| | 16 | 32 | 64 | 128 | 256 | 3.68 | 30 | 0.76 | 0.911 | 0.830 |
| | 8 | 16 | 32 | 64 | 128 | 1.42 | 23 | 0.24 | 0.892 | 0.803 |

where $\hat{y}_i$ and $y_i$ represent the $i$ pixel value of the predicted outcome and the target outcome, respectively, $\sum$ represents the summation of all pixel values, and $\varepsilon$ is a small normal number to avoid a denominator of zero. $BCE(\hat{y}, y)$ stands for BCE loss and is used to measure the difference between predicted result $\hat{y}$ and target result $y$ in dichotomous tasks. $Dice(\hat{y}, y)$ stands for Dice loss and is used to measure the similarity between the predicted result $\hat{y}$ and the target label $y$. This article improves the performance of the model on clinical diagnosis related datasets by combining a binary cross-entropy loss of 0.5× and Dice loss. The weight of 0.5 can be adjusted according to actual needs to balance the influence of the two in training.

## Analysis of experimental results

### Hyperparameter channel count analysis

RMIS-Net is a deep learning model for lesion boundary segmentation tasks in biomedical image analysis, and its performance is affected by the number of hyperparameter channels. In this article, the performance and computational overhead of the model are balanced by adjusting the values of the five channels C1 to C5.

Table 3 shows the performance comparison results of RMIS-Net in different channel configurations. It can be seen from the results that the higher the number of channels, the stronger the expression ability of the model RMIS-Net, so that the feature information in the image can be better learned. However, as the number of channels increases, the number of parameters, inference time, and GFLOPs of the model increase accordingly, resulting in increased computational overhead. Therefore, in practical applications, we need to consider the limitations of computing resources and the balance between performance and computing overhead. Although reducing the number of channels slightly reduces the performance of the model, the reduction is not drastic. Therefore, in order to train the network RMIS-Net more efficiently, all subsequent experiments in this article set the five channel values respectively: C1 = 16, C2 = 32, C3 = 64, C4 = 128, and C5 = 256. In practice, the appropriate channel count configuration can also be selected according to specific needs and computing resource constraints.

### Comparison of qualitative results of different network models

In order to more intuitively show that RMIS-Net can bring better segmentation results on the 2018 Data Science Bowl dataset and ISIC 2018 dataset, this article visualizes the segmentation results of RMIS-Net and other SOTA models. It can be seen from Fig. 10 that the RMIS-Net network proposed in this article is closer to the region correctly labeled by

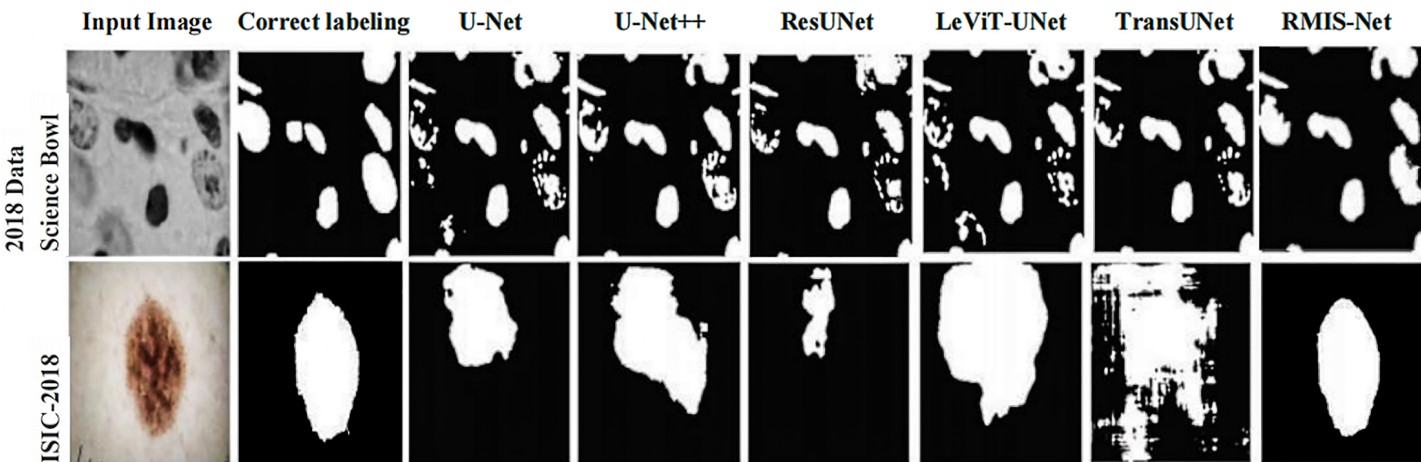

**Figure 10 Comparison of segmentation results of RMIS-Net and other SOTA models on two different datasets.**

**Table 4 Performance comparison of different network models on the 2018 Data Science Bowl dataset.**

| Model name | Model parameters (M) | Inference time (ms) | GFLOPs | Accuracy | Precision | Recall | F1-score | IoU |
|---|---|---|---|---|---|---|---|---|
| UNet (*Ronneberger, Fischer & Brox, 2015*) | 30.25 | 210 | 53.44 | 93.54% | 86.72% | 91.01% | 87.64% | 79.65% |
| UNet++ (*Zhou et al., 2018*) | 10.71 | 180 | 37.64 | 93.68% | 86.42% | 90.18% | 87.56% | 80.41% |
| ResUNet (*Zhang, Liu & Wang, 2018*) | 64.54 | 343 | 96.73 | 93.36% | 88.23% | 89.83% | 88.92% | 81.37% |
| LeViT-UNet (*Xu et al., 2023*) | 4.53 | 800 | 24.12 | 93.79% | 87.62% | 89.17% | 87.34% | 79.67% |
| TransUNet (*Chen et al., 2021*) | 99.61 | 263 | 40.56 | 93.18% | 89.74% | 89.21% | 89.96% | 81.08% |
| RMIS-Net | 3.68 | 30 | 0.76 | 93.90% | 90.12% | 90.37% | 91.12% | 82.93% |

the segmentation mask than other SOTA models, which proves that the RMIS-Net model proposed in this article has a better segmentation effect.

## *Comprehensive comparison of different network models*

(1) 2018 Data Science Bowl dataset

In order to further verify that the RMIS-Net network model proposed in this article can achieve better results than other SOTA models in the medical image segmentation task, this article uses multiple evaluation metrics to perform comprehensive experimental verification on the 2018 Data Science Bowl dataset. From the data in Table 4, it can be concluded that the model parameters, inference time (ms), computational complexity (GFLOPs), accuracy, precision, recall, F1-score and IoU of RMIS-Net reach 3.68 M, 30 ms, 0.76, 93.90%, 90.12%, 90.37%, 91.12% and 82.93%, respectively, indicating that RMIS-Net has obvious advantages in computing efficiency and performance metrics. The results show that RMIS-Net can perform better than other SOTA models in the nucleus segmentation task with lower model parameters, shorter inference time and less computational resources.

**Table 5 Performance comparison of different network models on ISIC-2018 datase.**

| Model name | Model parameters (M) | Inference time (ms) | GFLOPs | Accuracy | Precision | Recall | F1-score | IoU |
|---|---|---|---|---|---|---|---|---|
| UNet (*Ronneberger, Fischer & Brox, 2015*) | 31.75 | 233 | 56.34 | 93.63% | 86.84% | 91.37% | 86.34% | 82.12% |
| UNet++ (*Zhou et al., 2018*) | 9.86 | 212 | 40.71 | 93.87% | 86.63% | 90.45% | 87.93% | 80.36% |
| ResUNet (*Zhang, Liu & Wang, 2018*) | 67.34 | 354 | 97.63 | 93.57% | 88.79% | 89.63% | 87.45% | 81.37% |
| LeViT-UNet (*Xu et al., 2023*) | 7.63 | 836 | 26.34 | 93.26% | 87.12% | 89.72% | 87.37% | 80.46% |
| TransUNet (*Chen et al., 2021*) | 104.93 | 310 | 42.26 | 93.39% | 89.37% | 89.83% | 82.76% | 76.36% |
| RMIS-Net | 5.47 | 35 | 1.25 | 93.97% | 90.16% | 90.39% | 90.45% | 83.81% |

**Table 6 Ablation experimental results of different components.**

| Model name | Model parameters (M) | Inference time (ms) | GFLOPs | F1 | IoU |
|---|---|---|---|---|---|
| Original UNet (*Ronneberger, Fischer & Brox, 2015*) | 30.25 | 210 | 53.44 | 0.876 | 0.797 |
| Reduced UNet | 10.79 | 56 | 11.24 | 0.864 | 0.774 |
| Conv Stage | 1.26 | 12 | 0.54 | 0.813 | 0.711 |
| Conv Stage + Tok-MLP w/o PE | 3.65 | 23 | 0.76 | 0.886 | 0.801 |
| Conv Stage + Tok-MLP + PE | 3.68 | 24 | 0.76 | 0.891 | 0.816 |
| Conv Stage + Shifted Tok-MLP (W) + PE | 3.68 | 28 | 0.76 | 0.893 | 0.824 |
| Conv Stage + Shifted Tok-MLP (H) + PE | 3.68 | 28 | 0.76 | 0.892 | 0.821 |
| Conv Stage + Shifted Tok-MLP (H+W) + PE | 3.68 | 30 | 0.76 | 0.911 | 0.830 |

(2) ISIC 2018 dataset

Similarly, the RMIS-Net network model and other SOTA network models are applied to the ISIC 2018 dataset, and the experimental results are shown in Table 5. RMIS-Net has lower model parameters, inference time, and computational complexity compared to other models. In addition, RMIS-Net also showed high performance in accuracy, precision, recall, F1-score and IoU metrics, reaching 93.97%, 90.16%, 90.39%, 90.45% and 83.81%, respectively. The results show that RMIS-Net has high accuracy and reliability in lesion boundary segmentation tasks.

### Model component analysis

In this section, ablation experiments were performed on different components in the RMIS-Net model on the 2018 Data Science Bowl dataset, and the results are shown in Table 6. After comprehensive analysis of the ablation experimental results, the following key conclusions are drawn:

(1) The number of model parameters and computational complexity have an important impact on model performance. From the experimental results, it can be seen that compared with the Original UNet model, the Reduced UNet model reduces the number of model parameters and computational complexity by about 64% and 79%, respectively, but only slightly decreases in F1 and IoU metrics. This result shows that the model can maintain good segmentation performance while keeping computing resource consumption low.

(2) The introduction of Tok-MLP operation and position embedding has a positive impact on model performance. It can be seen from the experimental results that the Conv Stage + Tok-MLP + PE model improves F1 and IoU metrics by 2.6% and 5.1%, respectively, compared with the Conv Stage model, which proves that Tok-MLP operation and position embedding can improve the semantic modeling ability of the model and the understanding of different position information.

(3) The introduction of Shifted Tok-MLP operations can further improve model performance. It can be seen from the experimental results that the Conv Stage + Shifted Tok-MLP (H+W) + PE model improves the F1 and IoU metrics by 1.9% and 1.4%, respectively, compared with the Conv Stage + Tok-MLP + PE model, which shows that the Shifted Tok-MLP operation can enhance the modeling ability of the model on long-distance dependencies, thereby improving the performance of the model in the image segmentation task.

## CONCLUSION AND FUTURE WORKS

Based on the multilayer perceptron, this article proposes a fast medical image segmentation network RMIS-Net, which aims to solve the problems of high complexity and low efficiency in medical image segmentation, and provides an efficient and accurate solution for medical image analysis. The network structure includes convolutional layer and shift-based fully connected layer, and a tokenized multilayer perceptron is adopted in the latent space to reduce the number of parameters and computational complexity of the network, so that the network can capture the feature information in the image more accurately, thereby improving the segmentation accuracy of the model. Experimental results on the 2018 Data Science Bowl dataset and the ISIC 2018 Lesion Boundary Segmentation dataset show that the RMIS-Net network model proposed in this article can show better performance than other models in the medical segmentation task with lower model parameters, shorter inference time and less computing resources. In the future, the applicability of RMIS-Net on different types of medical image datasets will be further explored to verify its robustness and versatility in diverse clinical scenarios.

Although our proposed model has already achieved significant improvements in parameter count and computational complexity, its network architecture can still be further optimized. In the future, techniques such as self-attention mechanisms could be introduced to enhance the network's feature extraction and information propagation capabilities. Additionally, the model could be extended to multi-modal medical image segmentation tasks, such as fusing MRI and CT images for tumor segmentation. It could also be explored for real-time medical image segmentation tasks, such as real-time surgical navigation and real-time tumor segmentation, providing clinicians with real-time image analysis tools. Furthermore, the model could be applied to other medical image analysis tasks, such as object detection, image generation, and image reconstruction. Further optimization of the model's generalization performance and network efficiency should also be pursued.

### Funding

This article was supported by the Natural Science Foundation of Henan Province (Nos. 232300421354 and 232300420426), the Science and Technology Project of Henan Province (Nos. 222102110366 and 242102210045), and the Key Scientific Research Project of Colleges and Universities in Henan Province (No. 22A110014). There was no additional external funding received for this study. The funders had no role in study design, data collection and analysis, decision to publish, or preparation of the manuscript.

### Grant Disclosures

The following grant information was disclosed by the authors:
Natural Science Foundation of Henan Province: 232300421354, 232300420426.
Science and Technology Project of Henan Province: 222102110366, 242102210045.
Key Scientific Research Project of Colleges and Universities in Henan Province: 22A110014.

### Competing Interests

The authors declare that there are no conflict of interests regarding the publication of this article.

### Author Contributions

- Binbin Zhang conceived and designed the experiments, performed the computation work, authored or reviewed drafts of the article, and approved the final draft.
- Guoliang Xu analyzed the data, authored or reviewed drafts of the article, and approved the final draft.
- Yiying Xing performed the computation work, prepared figures and/or tables, authored or reviewed drafts of the article, and approved the final draft.
- Nanjie Li conceived and designed the experiments, performed the experiments, authored or reviewed drafts of the article, and approved the final draft.
- Deguang Li performed the computation work, authored or reviewed drafts of the article, and approved the final draft.

### Data Availability

The code is available at GitHub and Zenodo:

- https://github.com/binbinzhang1/RMISNet.
- binbinzhang1. (2024). binbinzhang1/RMISNet: v1.0.0 (v1.0.0). Zenodo. https://doi.org/10.5281/zenodo.12731207.

The third-party datasets are available at:

- https://www.kaggle.com/c/data-science-bowl-2018/.
- https://challenge.isic-archive.com/landing/2018/45/.

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
