# Peer review of "RMIS-Net: a fast medical image segmentation network based on multilayer perceptron"

_PeerJ Computer Science, doi:10.7717/peerj-cs.2882_

## Round 0.1 · original submission · Major Revisions

Based on the referee reports, I recommend a major revision of the manuscript. The author should improve the manuscript, taking carefully into account the comments of the reviewers in the reports and resubmit the paper.

Reviewer 1 ·

Basic reporting

- This manuscript presents a new model called RMIS-Net which is a Fast Medical Image Segmentation Network based on multi-layer perceptron.

- This manuscripts is well-written and easy to follow.

- Existing works are covered till 2022. Add few papers from 2023 to 2024 in this area of research.

- Improve the visibility of figure-3.

- Refer # 29 and 30 should be in footnotes.

- Add a paragraph in the last of the Introduction section which explains the flow of the rest of the paper.

- Add a sub-section in the beginning of Section-2 as "Medical image segmentation network based on Machine Learning Approaches" which cover literature based on traditional machine learning-based approaches.

- Add a sub-section as "Current Status and Limitation" in the Section-2 which highlight the overall crux of existing works. Here, you can explain why your work is different from others.

Experimental design

- Add a new recent benchmark dataset for evaluations. Both datasets are of 2018. It it not existing then create a new one for better evaluation of the RMIS-Net model.

- In tables 4 and 5, all baseline models should have references also.

- In table 6, report results in terms "accuracy" also. So, add a column for the same.

- Define all considered evaluation metrics in sub-section: 4.2 in terms of equations.

Validity of the findings

- Authors need to highlight novelty separately for which they have to create an new sub-section called "Our Contributions" in Section 1.

- Future works need to more detail as a separate section. Alternatively, rename: Conclusion as Conclusion and Future Works.

Additional comments

This manuscripts is self-explanatory, but needs revision in its present form.

Reviewer 2 ·

Basic reporting

- The article's English is quite weak.
- The purpose of the article is unclear, additions should be made to the introduction section accordingly.
- The contributions of the article are listed. However, what purpose do these contributions provide? The contributions section should be detailed instead of giving one sentence each.
- The organization of the paper should be given at the end of the introduction section.
- "Dice" or "DICE", a single usage form should be preferred.
- What does RMIS stand for, its expansion should be given where it is first used.
-Explanations should be written in Figure 6. Text should be added to understand what each module is expressed in the figure.

Experimental design

- How were the C1, C2,..., C5 channel number values ​​determined in the proposed method.
- Before moving on to the subheading in section 4, information about the experimental conditions should be given. Under which conditions and with which parameters was the experiment performed?
- Why were horizontal flipping, rotation, and cutting applied as data augmentation methods? Why were different methods not preferred?

Validity of the findings

- What is the data partitioning ratio for training, testing, and validation according to the values ​​in Table 1?
- More detailed information should be provided about the metrics used.
- Can the channel number values ​​(Table 3) be varied?
- The conclusion section should be expanded.

Additional comments

The article titled "RMIS-Net: A Fast Medical Image Segmentation Network Based on Multilayer Perceptron" should be revised according to the following points.

Reviewer 3 ·

Basic reporting

+ There are totally 30 references which include over 15 references from conferences level. I strongly recommend that authors should cite the same level (journal level) with this journal
+ Moreover, reference [29] and [30] are hyperlinks which have wrong citation format, please correct it
+ Introduction section seems to be long. Authors could reduce it, as well the description for main contribution should be located at the end of section 2 (Related works)
+ Usually, when citing the number of reference such as [10] or [8, 9], authors should use some general terms, for instance researchers, developers and so on
+ English needs to be improved

Experimental design

+ There is no description in Fig. 1, Fig. 6 such as what is input/output, what component in each stage
+ Fig. 2 is extended too much, some figures are blurred such as Fig. 7, Fig. 8 and Fig. 9
+ Mathematical equation should be deployed in the theoretical section
+ In the experimental setup, some components for hardware/software/firmware or platform must be explained

Validity of the findings

+ Authors did not clearly state their findings

---

## Round 0.2 · Major Revisions

Based on the referee reports, I recommend a major manuscript revision. The author should improve the manuscript, carefully consider the reviewers' comments in the reports, and resubmit the paper.

**Language Note:** The review process has identified that the English language must be improved. PeerJ can provide language editing services - please contact us at [email protected] for pricing (be sure to provide your manuscript number and title). Alternatively, you should make your own arrangements to improve the language quality and provide details in your response letter. – PeerJ Staff

Reviewer 1 ·

Basic reporting

Authors have addressed all comments.

Experimental design

Authors have addressed all comments.

Validity of the findings

Authors have addressed all comments.

Additional comments

Authors have addressed all comments. Just check typos and gramatical mistakes. For example, we introduces (last paragraph- Section: Introduction). Fixed it.

References in tables : 4, 5, and 6 are like superscript. So, make it align with model name.

For example,

UNet+ [5], ResUNet+ [8]

Reviewer 2 ·

Basic reporting

-

Experimental design

-

Validity of the findings

-

Additional comments

It is clear that the article has become more readable and understandable with the revision made. It can be accepted as it is.

Reviewer 3 ·

Basic reporting

+ English must be improved. Authors must use the Professional English Editing Service to modify. In the submission of revision, authors must submit the certificate of this service
+ Caption of image must be re-written. It is very short and not enough meaning to cover the purpose of figure
+ There are only six equations in the theoretical developments for the scientific publication. It means that authors have a few of scientific evident to prove your works

Experimental design

+ Source of dataset must be clearly specified in the body text
+ Every model has pros and cons, please produce a section for "Threat to Validity" to discuss the disadvantages and drawbacks

Validity of the findings

+ Before comparing to the other works, authors should conduct more experiments to confirm the proposed approach

---

## Round 0.3 · accepted · Accept

The author has addressed the reviewer comments properly. So, I think it's possible to publish the manuscript.

Reviewer 3 ·

Basic reporting

it can be published

Experimental design

ok

Validity of the findings

ok